# Ferroelectricity and Piezoelectricity in 2D Van der Waals CuInP_2_S_6_ Ferroelectric Tunnel Junctions

**DOI:** 10.3390/nano12152516

**Published:** 2022-07-22

**Authors:** Tingting Jia, Yanrong Chen, Yali Cai, Wenbin Dai, Chong Zhang, Liang Yu, Wenfeng Yue, Hideo Kimura, Yingbang Yao, Shuhui Yu, Quansheng Guo, Zhenxiang Cheng

**Affiliations:** 1Institute of Advanced Materials Science and Engineering, Shenzhen Institutes of Advanced Technology, Chinese Academy of Sciences, Shenzhen 518055, China; ry.chen@siat.ac.cn (Y.C.); yl.cai@siat.ac.cn (Y.C.); wb.dai@siat.ac.cn (W.D.); chong.zhang@siat.ac.cn (C.Z.); liang.yu@siat.ac.cn (L.Y.); wf.yue@siat.ac.cn (W.Y.); sh.yu@siat.ac.cn (S.Y.); 2School of Materials Science and Engineering, Hubei University, Wuhan 430062, China; 3School of Materials and Energy, Guangdong University of Technology, Guangzhou 510006, China; ybyao@gdut.edu.cn; 4Nano Science and Technology Institute, University of Science and Technology of China, Suzhou 215125, China; 5School of Environmental and Material Engineering, Yantai University, Yantai 264005, China; deokimura@yahoo.co.jp; 6Institute for Superconducting & Electronic Materials, University of Wollongong, Innovation Campus, North Wollongong, NSW 2500, Australia

**Keywords:** two-dimensional materials, ferroelectric properties, scanning probe microscope, negative piezoelectricity, phase segregation

## Abstract

CuInP2S6 (CIPS) is a novel two-dimensional (2D) van der Waals (vdW) ferroelectric layered material with a Curie temperature of *T*_C_~315 K, making it promising for great potential applications in electronic and photoelectric devices. Herein, the ferroelectric and electric properties of CIPS at different thicknesses are carefully evaluated by scanning probe microscopy techniques. Some defects in some local regions due to Cu deficiency lead to a CuInP2S6–In_4/3_P_2_S_6_ (CIPS–IPS) paraelectric phase coexisting with the CIPS ferroelectric phase. An electrochemical strain microscopy (ESM) study reveals that the relaxation times corresponding to the Cu ions and the IPS ionospheres are not the same, with a significant difference in their response to DC voltage, related to the rectification effect of the ferroelectric tunnel junction (FTJ). The electric properties of the FTJ indicate Cu^+^ ion migration and propose that the current flow and device performance are dynamically controlled by an interfacial Schottky barrier. The addition of the ferroelectricity of CIPS opens up applications in memories and sensors, actuators, and even spin-orbit devices based on 2D vdW heterostructures.

## 1. Introduction

Two-dimensional (2D) ferroelectric materials are gaining extensive attention. It can effectively improve device performance when applied to devices such as memory, capacitors, actuators, and sensors [1,2,3,4]. Nowadays, people regard such kinds of 2D materials as var der Waals (vdW) layered ferroelectric materials, which benefit from covalently bonded polar or non-polar monolayers by Van der Waals forces and exhibit ferroelectric properties. However, reports of ferroelectric 2D materials at room temperature are rare. Because of the depolarization field with decreasing thickness, there is an enormous challenge in maintaining ferroelectricity in ultrathin ferroelectric films. Ferroelectricity is remained elusive in the 2D material library [5,6] Van der Waals (vdW) layered ferroelectric materials has become a promising research branch in condensed matter physics [7,8], among which copper indium thiophosphate, CuInP_2_S_6_ (CIPS), is one of the most representative materials because of its room temperature ferroelectricity [9]. CIPS is promised to play an important role in nonvolatile memory. A recent experiment reported that the vdW ferroelectric tunnel junction (FTJ) device based on CIPS achieved a high tunneling electroresistance (TER) ratio [10,11]. Quantum transport device simulations of an FTJ based on CIPS and graphene demonstrate that scaling of the ferroelectric layer thickness exponentially not only significantly builds up the ferroelectric tunneling ON current but also reduces the read latency, in addition to enabling the FTJs with CIPS bilayers or trilayers to read speed in nanoseconds [12]. There are also outstanding endurance and retention characteristics for FTJ devices [3].

It was reported that giant intrinsic negative longitudinal piezoelectricity was observed in 2D layered CIPS. Lu You et al. [12] tested the converse piezoelectric effects of poly(vinylidene difluoride (PVDF), CIPS, and lead zirconate titanate (PZT) using a piezoelectric microscope and concluded that the electromechanical properties of CIPS are the same as those of PVDF with a negative longitudinal piezoelectric effect but opposite to those of PZT. This abnormal electromechanical phenomenon is caused by the significant deformation sensitivity of the weak interlayer interaction and is mediated by the high displacive instability of Cu ions. Several groups have discussed the origins of negative piezoelectricity. Yubo Qi and Andrew M. Pappe [13] attributed the negative piezoelectricity to the “lag of Wannier center” effect by proposing a negative clamped-ion term in the low-dimensional layered materials. John A. Brehm et al. [14] combined first-principles calculations with local electromechanical material characterization. They predicted and verified the existence of a uniaxial quadruple potential well for Cu displacements achieved by the van der Waals gap in CIPS. This led to the explanation that the negative longitudinal piezoelectric coefficient stems from the low polarization and very high sensitivity to a strain of the Cu atoms within the layer. Due to the negative piezoelectricity, CIPS has more complex piezoelectric properties, which gives it potential application prospects for 2D vdW materials with the same complex piezoelectric behavior in calculation and energy conversion. Therefore, it is crucial to study the piezoelectric behavior of CIPS.

Recently, to achieve resistance changes by ferroelectric switching, a ferroelectric field-effect transistor (FeFET) has been proposed, which uses ferroelectric materials instead of the oxide layer in a FET [15,16,17]. The causesFeFET exhibits two resistive states due to the hysteresis of ferroelectric switching. Furthermore, ferroelectric tunneling junctions and ferroelectric diodes were also investigated. It is reported that the polarization-modulation of Schottky-like barriers realizes the resistive change [18,19]. Yet, the vast majority of these devices use conventional ferroelectric materials such as PbTiO_3_ and BaTiO_3_ [20]. Affected by the three-dimensional nature of the ferroelectric oxide lattices, it is necessary for epitaxially grown high-quality films to select the substrates with a small lattice mismatch [21]. This seriously limits the possible application of materials in ferroelectric heterostructure devices. Therefore, it may be fundamentally and practically beneficial to study weakly bonded non-oxide ferroelectric compounds. Beyond that, the pioneering work on graphene has attracted an intense search for other 2D materials [22,23].

In this work, the crystal structure and ferroelectricity in crystalline CIPS nanoflakes are investigated at room temperature. Thin layer crystals in their pristine state show ferroelectric domains that are visualized directly. Piezoresponse force microscopy (PFM) measurements show that the polarization is stable and switchable in different layers of CIPS, and a negative piezoelectric effect is observed. Furthermore, we observed the ion migration phenomenon in CIPS via electrochemical strain microscopy (ESM). Finally, to obtain insight into the nature of the TER phenomena in the CIPS—based FTJs, we investigated their current—voltage (*I*–*V*) characteristics with a conductive probe microscope.

## 2. Experimental Section

### 2.1. Sample Preparation and Structural Characterization

All CIPS crystals were purchased from 6Carbon Technology (Shenzhen, China). The purchased synthetic bulk crystals were mechanically exfoliated onto platinized silicon substrates to obtain the flakes. The flake thickness was measured with a three-dimensional (3D) laser confocal microscope (VK-X1100, KEYENCE CORPORATION., Itasca, IL, USA) and an atomic force microscope (AFM, Cypher S, OXFORD INSTRUMENTS, Abingdon, Oxfordshire, UK). The microscope objective lenses used in our experiments are ×5, ×10, ×20, ×50, ×100, and the microscopic ocular is ×5. The numerical aperture and working distance of the microscope objective lens with ×100 magnification are 0.3 mm and 4.7 mm, respectively. The confocal Raman system (LabRAM HR, HORIBA Instuments(SHANGHAI) co. LTD, Shanghai, China) with 532 nm laser excitation was used to collect Raman spectra. The morphologies and structures of the as—prepared samples were collected by transmission electron microscopy (TEM, JEM-3200FS, JEOL (BEIJING) Co., Ltd. GUANGZHOU BRANCH, Beijing, China).

### 2.2. Piezoelectric Force Microscopy (PFM) Measurements

The PFM measurements were performed on μm—sized CIPS flakes on the crystal surface. The silver paste was used to attach the sample to the sample stage, which served as an electric back contact. PFM image scanning measurements were performed using a piezoelectric force microscope (PFM) with Au- and diamond-coated Si cantilever tips (FM-LC, 100 kHz and 8 N/m), respectively. The cantilever tip acted as a local variable electrode. The electric voltage was applied to the sample surface via the conductive PFM tip. The sample was cleaved using Scotch tape directly before the measurements. The topography of CIPS flakes was probed in AC mode, while the piezoelectric and ferroelectric responses were measured using dual AC resonance tracking PFM (DART-PFM) mode. The local piezoresponse hysteresis loops were measured 10 times at each position at multiple different arbitrary points.

### 2.3. Electrochemical Strain Microscopy (ESM) Measurements

Relaxation and variable ESM measurements were performed using an atomic force microscope. Conductive Au- and diamond-coated Si cantilever tips (FM-LC, 100 kHz, and 8 N/m) were used in all measurements.

### 2.4. Conductive Atomic Force Microscopy (C-AFM) Measurements

Local current-voltage and current phase diagram measurements were performed by C-AFM using with Au- and diamond-coated Si cantilever tips (FM-LC, 100 kHz, and 8 N/m). *I*–*V* curves were collected perpendicular to the CIPS samples in C-AFM mode.

## 3. Results and Discussion

### 3.1. Structural Characterization

The room-temperature crystal structure of CIPS was first discovered in 1995 using single-crystal X-ray diffraction [24]. Bulk CIPS is composed of vertically stacked, weakly interacting layers bound together by vdW interactions. The metal cations and P–P pairs fill the octahedral voids in the sulfur framework in a CIPS crystal (Figure 1a). A complete unit cell is reported to consist of two adjacent monolayers due to the site exchange between Cu and the P–P pairs from one layer to another [25]. Figure 1b shows two typical Raman spectra measured in different regions of a CIPS flake on a Pt/Si substrate. In general, we observed five Raman active modes, including δ(S–P–P) modes at 162.9 cm^−1^, δ(S–P–S) modes around 226.1 cm^−1,^ and 263.3 cm^−1^, an active mode from the cations at 303.2 cm^−1^, andυ(P–P) and υ(P–S) modes at 374.3 cm^−1^ and 436.4 cm^−1^, respectively. The tested Raman results are consistent with the former reports on CIPS crystals [26,27]. P–S bond stretching vibrations (υ) can be resolved to A_1g_ + A_2u_ + E_u_ + E_g_. The S–P–S modes change the S–P–S angles, resulting in another set of A_1g_ + A_2u_ + E_u_ + E_g_. The P–P bond is associated with bending the PS_3_ groups, which accounts for E_u_ + E_g_. Twisting (P_2_S_6_)^4−^ oscillation is related to the A_1u_ mode [27]. The Raman spectra in orange showed a strong peak at ~303 cm^−1^, which corresponds to the cations [26]. The blue curve showed a weak and broad peak at ~303 cm^−1^, which is mainly related to the reduction of Cu content. It is believed to originate from the paraelectric phase of CIPS–IPS, where IPS stands for In_4/3_P_2_S_6_, appearing in the flake due to Cu^+^ deficiency [26,27]. The transmission electron microscope (TEM) images show two structures on the same CIPS flakes (Figure 1c), which correspond to the monoclinic and triclinic phases, consistent with the Raman spectrum.

### 3.2. Ferroelectric Domain and Domain Switching

To approve the ferroelectricity of CIPS flakes with different thicknesses, the domain distribution of the CIPS flakes was investigated in 3 nm, 9 nm, 12 nm, 21 nm, 35 nm, and 78 nm CIPS flakes by piezoresponse force microscopy (PFM). The domain evolution of the CIPS flakes was observed with height, amplitude, and phase images, as shown in Figure 2a–f. The domain size varied with the increasing thickness of the CIPS. As shown in Figure 2a–f, the size of the ferroelectric domains increases with the thickness of the CIPS flakes. Domains vary in size but are on the order of ~100 nm and ~1 μm in diameter, respectively, different from reports in the literature [7]. In our experience, it is easy to observe clear domain structures in CIPS flakes with thicknesses in the thickness range from 20 to 40 nm, as shown in Figure 2d,e. We observed the reversed piezoelectric effect in the typical amplitude-voltage “butterfly” loops by scanning the piezoresponse hysteresis loops in dual AC resonance tracking PFM (DART-PFM) mode. There are regions where the ferroelectric inversion loop disappears in the 20 nm CIPS flakes, which may be related to the paraelectric phase caused by Cu^2+^ segregation.

According to the above results, we chose the CIPS sample with a thickness of 20 nm to investigate the ferroelectric switching behavior. Well-defined butterfly loops of the saturated PFM amplitude and the distinct 180° switching of the phase signals were observed, as shown in Figure 3a,b, indicating switchable ferroelectric polarization in the CIPS flakes with different thicknesses (28 nm and 20 nm). We divided the hysteresis loops into four sections, as shown in Figure 3a–c, and observed that the direction of the polarization switching of CIPS was consistent with that of PVDF [28], showing negative piezoelectricity, which arose from the low polarization of Cu atoms in the CIPS layer [14]. We still observed abnormal butterflies and switching of the phase loops in the 25 nm and 20 nm CIPS, however, as shown in Figure 3c,d. There are the following several possible reasons for the abnormal switching loops: One, when sweeping the electric fields in the reverse sequence, the piezoelectric response curves show asymmetric shapes. This phenomenon was attributed to the existence of remnant-injected electrons. Due to the different diffusion distances in both the positive and negative electric range, the injected electrons cannot be eliminated absolutely when applying a lower electric field [29]. Two, the abnormal hysteresis loops may also relate to negative electrostriction and electrostatic signal contributions accompanied by charge injection during scanning [30]. Three, abnormal domain switching is generated due to in-plane ionic migration in CIPS [31]. Since the Cu is deficient, when applying an electric field, a few regions of the CIPS undergo a chemical phase separation into a paraelectric In_4/3_P_2_S_6_ phase and a ferroelectric CuInP_2_S_6_ phase [32]. Abnormal hysteresis loops may be from the local paraelectric phase of the CuInP_2_S_6_–In_4/3_P_2_S_6_ region due to Cu deficiency [27].

To investigate the in-plane ionic migration, we carried out a relaxation test to analyze the local ionic dynamics of CIPS via electrochemical strain measurements (ESM), as shown in Figure 4 [33,34]. As schematically shown in Figure 4a, a direct-current (DC) voltage is applied on top of an AC voltage to induce a longer-range redistribution of ions in CIPS flakes during the probe. After removing the DC voltage, the ions relax back to their original equilibrium state. The local dynamics can be deduced from the time constant associated with the relaxation of ESM amplitude, as shown in Figure 4b. The DC voltage is applied on top of the AC following the profile shown in Figure 4a to induce a longer-range redistribution of Cu^+^ ions; meanwhile, polarizing the sample over a larger scale. The faster relaxation shown in Figure 4d corresponds to the Vegard strain directly related to the ionic concentration of Cu^+^. While slower relaxation in Figure 4d is because of the induced electrochemical dipoles and results from the readjustment of the negative InP_2_S_6_^−^ in response to the redistributed Cu^+^. The process takes longer over a much shorter distance. This phenomenon verifies our conjecture that the IPS paraelectric phase forms in the Cu-absent region due to the Cu ion migration, with no ferroelectric polarization switching. That is, the relaxation of Cu ions is associated with the change in ferroelectric polarization after the addition of voltage, and the longer time is related to the saturation phenomenon of polarization specific to CIPS reported in the literature.

### 3.3. Electric Properties of Pt/CuInP_2_S_6_/Au FTJ

CIPS by itself is the only two-dimensional ferroelectric material with a ferroelectric transition temperature (*T_c_*) just over room temperature. Based on the above studies, we have observed the paraelectric phase in CIPS flakes, but it is rarely reported whether the paraelectric phase affects the device’s performance. In this work, we used conductive AFM to study the electric properties of a CIPS FTJ as shown in Figure 5. The electrical characterization of an Au/CIPS/Pt vdW FTJ is shown in Figure 5a for a device with a 2-nm-thick CIPS layer. Figure 5b,c shows the topography and current images of the CIPS flakes. The leakage current scanning was performed within a 3 × 3 μm^2^ area at a read voltage of 10 mV. The observed local conductive path regions indicated good electrical conductivity in the CIPS flakes.

Figure 5d presents the *I*–*V* curves of a CIPS FTJ, measured with varying sweep ranges (*V*_max_ from 2 V to 3.5 V). We can observe resistive switching in both positive and negative voltage ranges, demonstrating that FTJ has superior continuous current modulation and self-rectification functions. The *I*–*V* curves show a nonsymmetrical contour, and worse symmetry appears with increased voltage. This corresponds to the Cu^2+^ migration process. Before the Cu^2+^ migration, a Schottky barrier must be overcome. Due to the difference in the work functions between Au and Pt, the current of the *I*–*V* curve under a positive voltage and a negative voltage is asymmetric. The current limiting behavior in the negative range is similar to the rectifying effect of diodes. As the applied voltage increases, the *I*–*V* curves show more obvious hysteresis in the positive range, which is always regarded as a resistance switching behavior. As shown in Figure 5e, the I_on_ and I_off_ correspond to the on-current and off-current when applying different voltages during the resistance switching. As shown in the inset of Figure 5e, the *I_on_/I_off_* of the FTJ with an ultra-thin CIPS film is over 200, which is comparable with the previous results [10], indicating that CIPS has good development prospects in the research on and application of nonvolatile memory devices.

## 4. Conclusions

In summary, we clearly observed the domain structure of CIPS with a thickness of 20–40 nm in PFM measurements and found that the hysteresis loop of CIPS showed the same negative longitudinal piezoelectric effect as PVDF, indicating the complexity of the piezoelectric response of CIPS. We used electrochemical strain microscopy (ESM) to further study Cu^2+^ segregation and discovered different relaxation times between the Cu^2+^ and an InP_2_S_6_^2−^ ion groups, with both ions showing significant differences in their DC voltage response. These observations imply that in the regions where the ferroelectric inversion loop cannot be observed, Cu^2+^ segregation occurs, and InP_2_S_6_^2−^ paraelectric phase is formed. In addition, the Cu^2+^ relaxation is related to the change in the ferroelectric polarization after the application of voltage, and the longer relaxation time is relevant to the distinctive ferroelectric polarization saturation in CIPS^2^. Finally, we constructed a Pt/CIPS/Au FTJ. We observed resistance switches in the positive and negative voltage ranges, demonstrating that FTJ has superior continuous current modulation and self-rectification functions. Combined with its piezoelectric characteristics, layered ferroelectric CIPS is suitable not only for memory, but also for sensors, actuators, and even spin-orbit devices based on vdW heterostructures.

## Figures and Tables

**Figure 1 nanomaterials-12-02516-f001:**
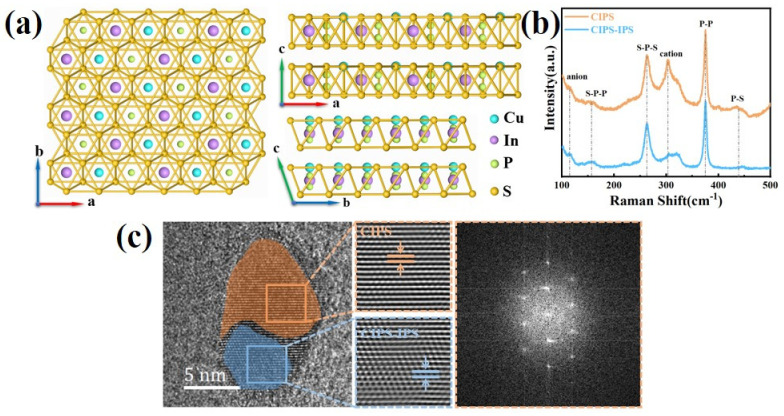
Material characterization of CIPS flake. (**a**) Top and side views of the CIPS crystal structure. In the atomic model, the yellow networks are S triangular networks, and the green, purple, and blue balls are P, In, and Cu atoms, respectively. (**b**) Raman spectra of CIPS flakes, including ferroelectric and paraelectric phases on the Pt substrate with 532 nm laser excitation. (**c**) The TEM characterizations of CIPS crystal include fast Fourier transform (FFT, **right**) and filtered inverse FFT (**middle**) patterns of the selected areas, respectively.

**Figure 2 nanomaterials-12-02516-f002:**
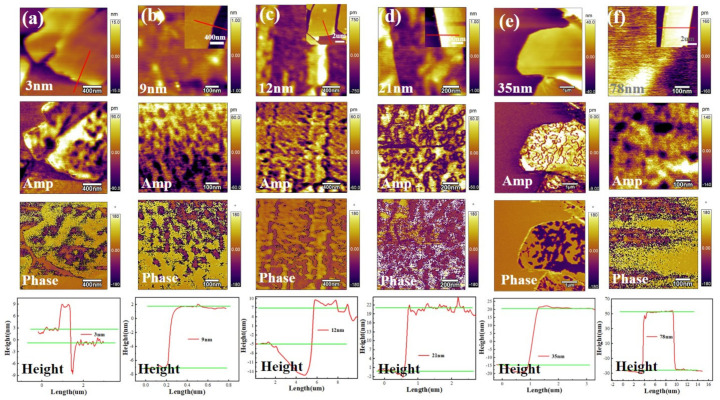
PFM images including height, amplitude, and phase image (upper three), and height curves (bottom row) of CIPS flakes with different thicknesses. (**a**) 3 nm, (**b**) 9 nm, (**c**) 12 nm, (**d**) 21 nm, (**e**) 35 nm, and (**f**) 78 nm.

**Figure 3 nanomaterials-12-02516-f003:**
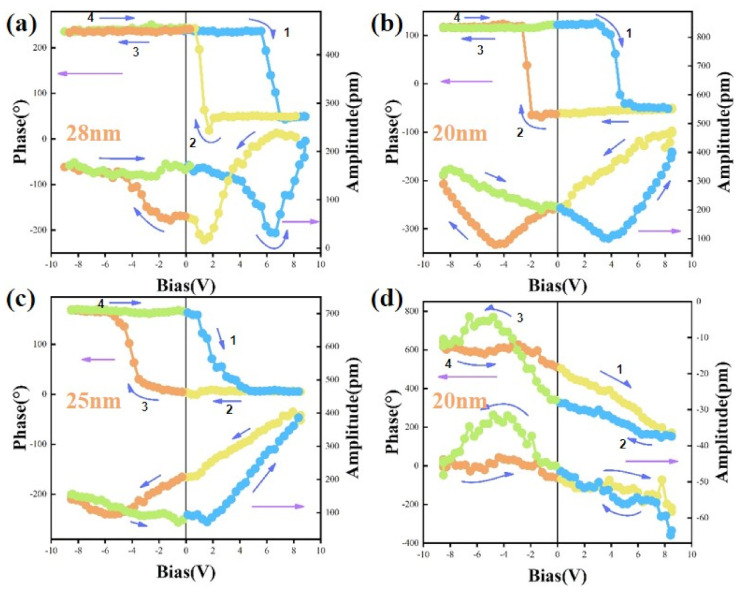
Ferroelectric polarization switching by PFM for CIPS flakes with different thicknesses. The PFM amplitude (green) and phase (blue) hysteresis loops during the switching process for CIPS flakes with thickness of (**a**) 28 nm, (**b**) 20 nm, (**c**) 25 nm, and (**d**) another area of the sample with the thickness of 20 nm.

**Figure 4 nanomaterials-12-02516-f004:**
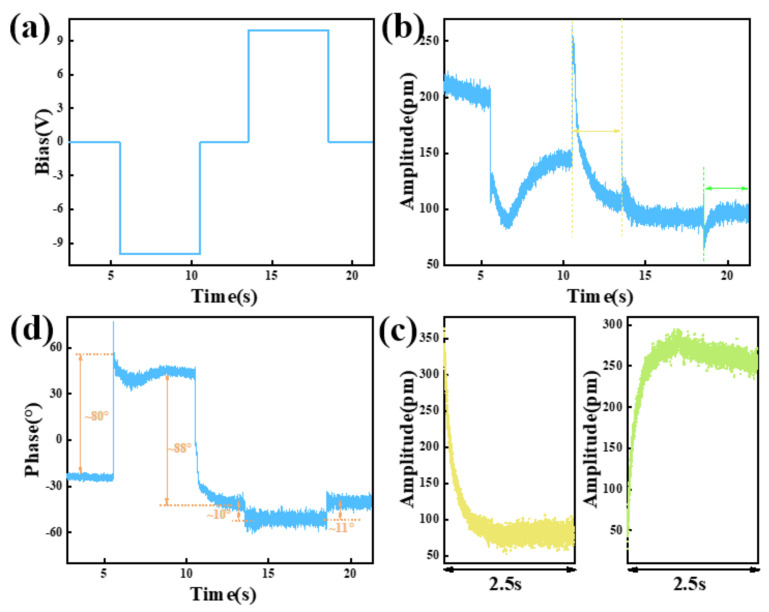
Relaxation dynamics in local electrochemical strain measurements of CIPS. (**a**) Illustrated DC profile applying in the relaxation measurements. (**b**,**d**) ESM amplitude-time and phase-time curves were obtained corresponding to the DC profile. (**c**) Zoomed—in relaxation curves of (**b**) were recorded after removing negative and positive DC voltage, respectively.

**Figure 5 nanomaterials-12-02516-f005:**
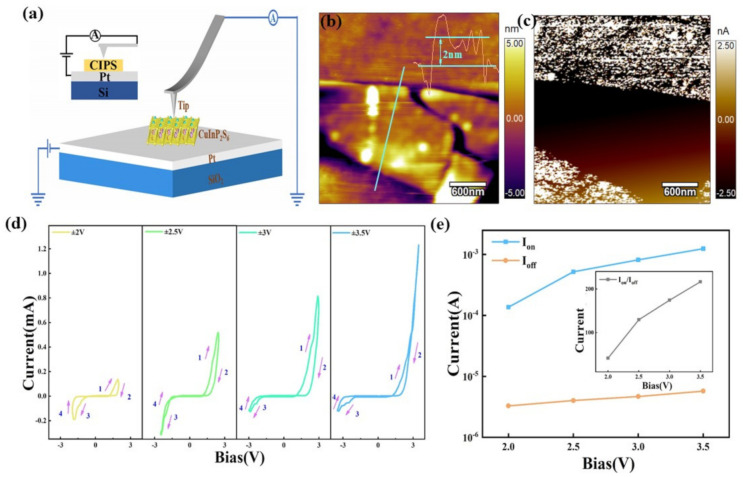
Electrical characterization of a Si/Pt/CIPS/Au diode with1.7 nm CIPS. (**a**) Schematic representation of the experimental setup for C-AFM measurements. (**b**) Topography image of the CIPS nano flake with a thickness of 2 nm on the Si/SiO_2_/Ti/Pt substrate. The inset shows the height map of the CIPS flake. (**c**) The corresponding current phase diagram of CIPS flakes. (**d**) *I*–*V* curves measured with increasing sweep voltages, where *V_Max_* is from 2 to 3.5 V. (**e**) *I_on_* and *I_off_* are the on and off current of the FTJ with resistance switching behavior, which are also corresponding to the low and high resistance states, read from (**d**) under different scanning voltages, with the inset the calculated switching ratio based on (**d**).

## Data Availability

Not applicable.

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
