# Peer review of "Ferroelectricity and Piezoelectricity in 2D Van der Waals CuInP2S6 Ferroelectric Tunnel Junctions"

_nanomaterials, 2022, doi:10.3390/nano12152516_

Round 1

Reviewer 1 Report

The paper refers to ferroelectric as well as electron transport properties of CuInP_2S_6 (CIPS) 2D nanoscale flakes. The Authors presented broad microstructural studies that well justify the conclusions related to ferroelectric domain structure as well as the segregation of Cu+ ions effect. Also interesting are transport properties of the tunneling junction device based on the CIPS compound. The presented I-V curves reveal the occurrence of the resistivity switching effect. Generally, the subject and results are very interesting and indicate a potential for electronic device applications. The aim, presentation and conclusions are clear and meaning-worthy. Therefore, I can recommend the paper for publication but some minor revision, according to listed below comments, is necessary.

1. Is it possible to show a map of regions in which the hysteresis (phase AFM) occurs and not occurs? In this way, it would be possible to directly show the Cu+ segregation.

2. It would be better, for potential readers, to precise define the I_on and I_off currents determined from the I-V curves.

3. I_on/I_off is unit-less. Remove (A) from the inset in Fig.5e.

4. Did you measured the I-V curves in different frequency? Potential readers, including me, could be interested in this relation.

Author Response

Dear Sir/Madam,

We appreciate your effort in reviewing our manuscript. We are also grateful for your suggestion and comments. We have revised the manuscript following your advice.

  1. Is it possible to show a map of regions in which the hysteresis (phase AFM) occurs and not occurs? In this way, it would be possible to directly show the Cu+ segregation.

RESPONSE

The hysteresis curve was obtained by SSPFM. Because the Cu+ segregation is radom and always occurs in a small region, it is not accessible to map the Cu+segregation directly.

  1. It would be better, for potential readers, to precise define the I_on and I_off currents determined from the I-V curves.

RESPONSE

The define of the Ion and Ioff is added in the revised manuscript. Thank you for the advice.

  1. Ion/Ioff is unit-less. Remove (A) from the inset in Fig.5e. 4. Did you measured the I-V curves in different frequency? Potential readers, including me, could be interested in this relation

RESPONSE

Sorry for the careless mistakes. We have revised Fig.5e. Since the I-V curve was obtained in the CAFM mode by applying a DC bias, we haven’t measured the frequency dependence. Since the due date is short for revision, we may carry out the measurement in our following work in the future.

Reviewer 2 Report

The manuscript nanomaterials-1824650 “Ferroelectricity and piezoelectricity in 2D vdW CuInP2S6 Ferroe-lectric tunnel junctions” by Tingting Jia, Yanrong Chen, Yali Cai, Wenbin Dai, Chong Zhang, Liang Yu, Wenfeng Yue, Hideo Kimura, Yingbang Yao, Shuhui Yu, Quansheng Guo, Zhenxiang Cheng report the investigation of the crystal structure and ferroelectricity in crystalline CIPS nanoflakes. The domain structure of CIPS was observed and described. Authors found that in the regions where the ferroelectric inversion loop cannot be observed, Cu2+ segregation occurs, and InP2S62- paraelectric phase is formed. The manuscript is well structures and clearly written. All conclusions are supported by comprehensive investigations. Therefore, I recommend a manuscript for publication after some minor revision which should improve the article. My remarks are follow:

- The abbreviation “ vdW” in the title of the manuscript is not a god idea, since not everyone will be able to understand what the authors had in mind. I recommend that you consider replacing to the full form “van der Waals”

- Please, provide more detail about microscopy used, such as microscope objective lens used, the numerical aperture and working distance.

- Breef description what is a Van der Waals layered ferroelectric materials is needed.

- It is not clear from the text of the article why the studied material is suitable for sensors, actuators, and even spin-orbital devices. The justification for these statements should be in the article's text, and not just in the conclusions.

Author Response

Dear Sir/Madam,

We appreciate your effort in reviewing our manuscript. We are also grateful for your suggestion and comments. We have revised the manuscript following your advice.

  1. The abbreviation “ vdW” in the title of the manuscript is not a god idea, since not everyone will be able to understand what the authors had in mind. I recommend that you consider replacing to the full form “van der Waals”

RESPONSE

We revised the title with the full form of “van der Waals” in the revision.

  1. Please, provide more detail about microscopy used, such as microscope objective lens used, the numerical aperture and working distance.

RESPONSE

We have inform the type of instrument:The 3d laser confocal microscope used in this work is the VK-X1100. However, we didn’t show detailes of the microscope during observing the materials. We add the information in the Experimental Section of the revision.

  1. Breef description what is a Van der Waals layered ferroelectric materials is needed.

RESPONSE

We add the description of “a Van der Waals layered ferroelectric materials” in the Introduction of the revision.

  1. It is not clear from the text of the article why the studied material is suitable for sensors, actuators, and even spin-orbital devices. The justification for these statements should be in the article's text, and not just in the conclusions.

RESPONSE

We also revised the manuscript and added descriptions in the Introdution and Results and Discussion sections.

Reviewer 3 Report

Ferroelectricity and piezoelectricity in 2D vdW CuInP2S6 Ferro-electric tunnel junctions

Recommendation: Accept after minor revision

In this paper, the authors evaluate the ferroelectric and electric properties of CIPS of different thicknesses by scanning probe microscopy techniques. Electrochemical strain microscopy (ESM) study reveals different relaxation times for Cu ions and the IPS ionospheres with a significant difference in their response to DC voltage. The authors observe a paraelectric phase and propose modulation of an interfacial Schottky barrier by Cu+ ion migration to explain the electric properties of the FTJ device.

The authors should consider the following comments before publication:

1.    The authors need to compare the FTJ performance against literature.

2.    The authors need to measure the retention and endurance of these FTJs to evaluate their performance as NVMs.

3.    What is the switching speed of these FTJ devices for memory applications?

Author Response

Dear Sir/Madam,

We appreciate your effort in reviewing our manuscript. We are also grateful for your suggestion and comments. We have revised the manuscript following your advice.

  1. The authors need to compare the FTJ performance against literature.

RESPONSE

We add the comparation in the results and discussion part in the revision.

  1. The authors need to measure the retention and endurance of these FTJs to evaluate their performance as NVMs.

RESPONSE

It is a good idea to measure the retention and endurance properties because good retention properties of FTJs are crucial for the memory application. However, it requests that a write 100 ns pulse of variable amplitude be applied to the device. We don’t have such an accessory to write pulse voltage with a pulse width in ns. Since the journal asked us to submit the revision in 5 days. We cannot get the retention data in a short time. Anyway, we will try our best to do the retention measurement, and we hope we can publish the data in the future.

  1. What is the switching speed of these FTJ devices for memory applications?

RESPONSE

It is a good question to consider the switching speed of FTJ devices for memory applications. Ma Chao et al. reported a high performance memristor based on a Ag/BaTiO3/Nb:SrTiO3 ferroelectric tunnel junction (FTJ) with the fastest operation speed (600 ps) and the highest number of states (32 states or 5 bits) per cell among the reported FTJs. (Nature Communications 2020,11, 1439. DOI: 10.1038/s41467-020-15249-1) There is still some FTJs show the switching speed at the order of magnitude of a hundred nanoseconds(ns). However, it is necessary to investigate the CMOS compatibility, scalability, and switching dynamics in an integrated FTJ device except for the switching speed. The performance of the memory device based on the CIPS will be discussed in our future work.